



# Modelling and assessment of urban flood hazards based on rainfall intensity-duration-frequency curves reformation

Reza Ghazavi[1*], Ali Moafi Rabori[2] and Mohsen Ahadnejad Reveshty[3]

[1.] *Associate Professor, Department of Watershed Management, Faculty of Natural*
*Resources and Earth Sciences, University of Kashan, Kashan, Iran*

[2.] *PhD Candidate of Watershed Management, Department of Watershed Management, Faculty of Natural Resources and Earth Sciences, University of Kashan, Kashan, Iran*

[3.] *Associate Professor, Department of Geography, Faculty of Humanities, University*
*of Zanjan, Iran*

*Corresponding author: ghazavi@kashanu.ac.ir, Department of Watershed Management, Faculty of Natural Resources and Earth Sciences, University of Kashan, Kashan, Iran*

[1]*ghazavi@kashanu.ac.ir;*[2]*ali.moafi@ut.ac.ir;*[3]*ahadnejad@gmail.com*

## Abstract

Estimate design storm based on rainfall intensity–duration–frequency (IDF) curves is an important parameter for hydrologic planning of urban areas. The main aim of this study was to estimate rainfall intensities of Zanjan city watershed based on overall relationship of rainfall IDF curves and appropriate model of hourly rainfall estimation 20 (Sherman method, Ghahreman and Abkhezr method). Hydrologic and hydraulic impacts of rainfall IDF curves change in flood properties was evaluated via Stormwater Management Model (SWMM). The accuracy of model simulations was confirmed based on the results of calibration. Design hyetographs in different return periods show that estimated rainfall depth via Sherman method are greater than 25 other method except for 2-year return period. According to Ghahreman and Abkhezr method, decrease of runoff peak was 30, 39, 41 and 42 percent for 5-10-20 and 50-year return periods respectively, while runoff peak for 2-year return period was increased by 20 percent.

**Key words:** Design storm, Drainage system, Flood, Rainfall IDF curve, Stormwater,
SWMM, Urban, Zanjan city

## Introduction

Storm water runoff management is a complex task in urban area due to variation and complexity of land use, population and social economic activities (Choi and Ball,



2002; Hoang et al., 2016). This issue will become more complex due to urban

development. By 2030, the urban population will reach 5 billion or 60 percent of the world's population (UN, 2006). In many countries, less than 5 percent of land occupied via urban area, consequently, concentration of human activities, shortages and unavailability of resources intensifies local competition for all types of resources, with water amongst the most vital (Zoppou, 2001). Urbanization increase flood risk in

urban areas due to local change in hydrological cycle and hydro-meteorological conditions (Huong and Pathirana, 2013; Ahilan et al., 2016; Eunsek et al., 2016). In an ideal situation, urban drainage systems would be planned, designed and analyzed with catchment modelling systems which fully replicated the important processes involved with the generation and transmission of stormwater (Choi and

Ball, 2002). Estimate and collection of input parameters (measured and inferred) is very important in the catchment modelling. Measured data such as catchment areas, pipe diameters, and depth or intensity of rainfall should physically measure, while inferred data are determined from the application of a model. Rainfall intensity-duration-frequency (IDF) curves is an important parameter that need to

measurement. Today, due to urban development, urban flooding have increased in terms of intensity and frequency. Various tools such as structures or non- structures was used for urban flooding management and control. Estimating the design storm has an important role in designing and operations of structural hard-engineered solutions for urban runoff management. Use of rainfall IDF curves is a critical method

for estimating the design storm. Change in rainfall condition lead to change in rainfall IDF curves. Sometimes rainfall IDF curves prepared via old data and need to update or reformation. A number of studies have focused on urban flood characteristics estimating based on rainfall IDF curves update. The effects of climate change on urban stormwater infrastructure was estimated in Canada (Watt et al., 2003). Based on

the results of this study, drainage systems with overflow capacity was introduced. Urban runoff in Esfahan city in Iran was estimated using maximum intensity of precipitation for 15-360 minute rainfall duration and based on the rational method determined the flooding urban area (Asgari et al., 2005). In order to estimate the runoff peaks and volumes for designing the urban drainage system in Quebec, the

impact of current and future climate change on the rainfall IDF curves and urban design storms was estimated using SWMM model (Desramaut, 2008). Modeling the impacts of urbanization and climate change on flood properties show that increase in



rainfall intensity and impervious surfaces will cause flashier runoff periods, greater peak flows and heightened risk of flooding (Semadeni-Davies et al., 2008). For developing

rainfall intensity-duration-frequency curves in scarce data region, in North-West of Angola, an index flood procedure was used for generate the theoretical regional distribution equation. The proposed Index reflects rainfall and runoff characteristics of each region (Ayman et al., 2011). Changes in flood frequencies of sewer systems and overflow frequencies of storage facilities have been quantified based on the climate

scenarios and related changes in rainfall statistics (Willems, 2011). In order to design the drainage structures, Ibrahim (2012) conducted a study for developing rainfall IDF relationship for two regions in Saudi Arabia. Using improved IDF relations in Khorasan region of Iran, a study was performed to determine the spatial distribution of storms (Akbari et al., 2014). During the last century, the concentration of

greenhouse gases has increased due to urban development and increasing of industrial activities (Prodanovic and Simonovic, 2007). This change can lead to changes in temperature and precipitation characteristics. The changes in precipitation characteristics should change rainfall IDF curves (Prodanovic and Simonovic, 2007). Change in IDF curve in urban area should lead to changes in

urban flood characteristics. This change is more pronounce in arid and semi-arid area such as Iran. The main aim of this study was to update rainfall IDF curves of the study area, to investigate the effects of rainfall intensity-duration-frequency curves reformation on urban flood characteristics using SWMM model, and to investigate the effects of rainfall IDF curves updating on peak and volume of flood.

## Study area

The study area is located in the center of Zanjan province, north-west of Iran (latitude 36° 38′ 26″ and 36° 42′ 20″N, longitude 48° 26′ 29″ and 48° 35′ 02″ E). Altitude of the study area ranging from 1590 m above mean sea level in the southern plain to 1773m in the northern mountain (Figure 1).  Total area of the study urban

watershed is about 39 km$^2$ and the mean annual rainfall is 290 mm. The main part of rainfall in the study area was occurred in the autumn and spring. Artificial canals draining the urban area and playing an important role in flood routing during storm events. Flow direction of this canals is from north to south of urban area and falling into Zanjanrood River. Gavazaang earth dam has been built at the north of the city.

This dam limit upstream surface water and floods, so flow of upstream watershed



don't arrive into the city. The study area includes central business district of zanjan city, public parks, green space, residences and streets. This city experienced rapid development and population expansion during 1956-2012.

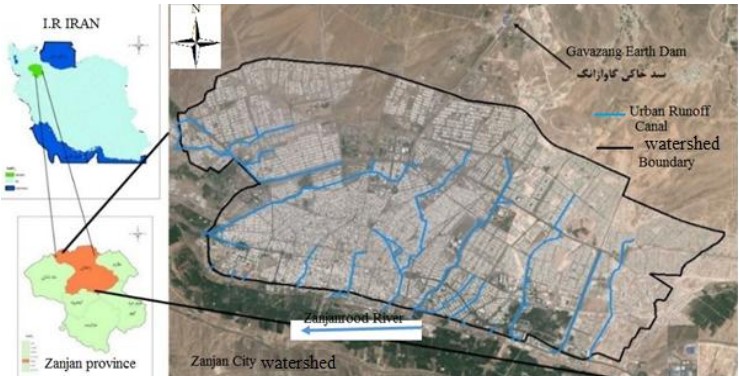

**Figure 1**. Locations of the Zanjan City Watershed

## Material and Methods

## Model description

The EPA Storm Water Management Model (SWMM) developed under the support of the US Environmental Protection Agency (Huber and Dickinson, 1992). SWMM is a
dynamic rainfall-runoff simulation model that computes runoff quantity and quality from primarily urban areas. SWMM widely used throughout the world for planning, analysis, and designing related to stormwater runoff, combined sewers, sanitary sewers, and other drainage systems in urban and non-urban areas. The runoff component of SWMM operates on a collection of sub catchment areas that receive
precipitation and generate runoff and pollutant loads. The routing portion of SWMM, transports this runoff through a system of pipes, channels, storage/treatment devices, pumps and regulators. SWMM tracks the quantity and quality of runoff generated within each sub catchment and the flow rate, flow depth, and quality of water in each pipe and channel during a simulation period comprised of multiple time
steps (Gironas et al., 2009).

## Model implementation

The primary objective of this study was to evaluate the hydrologic and hydraulic response of the study watershed to the rainfall IDF curves updating based on
increasing of the statistical period length of the rainfall data. Based on the rainfall


data of zanjan station (1972-1993), the rainfall IDF curves of zanjan station were prepared in 1995 using Sherman equation. The rainfall IDF curves of Iran were updated by Ghahreman and Abkhezr (2004), via increasing of the statistical period length (1972-2004). They illustrated a new general relationship for rainfall IDF curves

of Iran and indicated that previous relationships is not useful for estimating the 10-year hourly rainfall. In this study, in order to investigate the effect of rainfall IDF curves updating on the urban runoff peak and total volume of runoff, the rainfall hyetographs are prepared based on the both Sherman and Ghahreman and Abkhezr equations. Thus, SWMM model was executed using two design storm hyetographs.

The implementation of SWMM model necessitates several steps include; (1) identification of sub watersheds (2) representation of the channel network and (3) identification of the model parameters (Camorani et al., 2005).

## Identification of sub watershed

Sub watershed was identified based on urban drainage system. Basin boundary and sub watershed borders has been determined using land use maps, topographic map (1/2000), building blocks, direction of flow in canals and land survey. 64 sub watershed were determined (Figure 2 and Table 1).

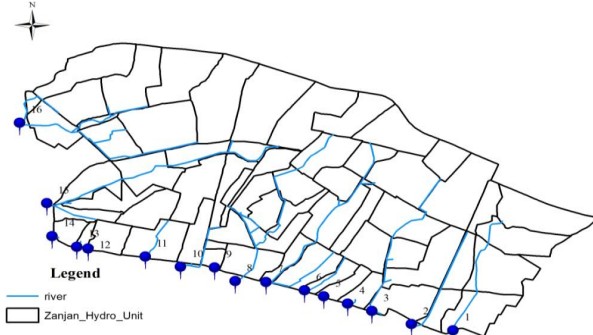

**Figure 2.** Urban drainage network, sub watershed and location of the watershed outlets of the study city

**Table 1.** Canals name, length and corresponding sub watershed area

| Canal name | 1 | 2 | 3 | 4 | 5 | 6 | 7 | 8 | 9 | 10 | 11 | 12 | 13 | 14 | 15 | 16 |
|---|---|---|---|---|---|---|---|---|---|---|---|---|---|---|---|---|
| Main Canal length (km) | 2.9 | 2.4 | 3.6 | 0.2 | 0.2 | 1.1 | 4.3 | 4 | 0.03 | 1.5 | 1.0 | 0.2 | 0.2 | 0.1 | 5.1 | 2.8 |
| Sub watershed area(km$^2$) | 3.7 | 1.1 | 4.6 | 0.3 | 0.3 | 0.9 | 4.9 | 4.5 | 0.2 | 1.5 | 1.6 | 0.4 | 0.1 | 0.3 | 6.3 | 7.9 |



## Urban drainage system representation

The canal-network as a link-node system were entered into the model. Additional nodes (junctions) has been inserted where a quick changes in links (conduits) characteristic was detected (such as change in geometry (depth, width), bed slope, roughness coefficient and shape) or when tributary canals is connected to the main canal. The network geometry (canal profile and cross-sections) has been derived

from the topographical map and land survey.

## Model parameters

Surface area, Manning roughness coefficient of canals, impervious and pervious area, average width of overland path, average surface slope, percent of impervious

area, depth of depression storage on impervious and pervious area, percent of impervious area with no depression storage and  infiltration parameters are prepared for implementation in SWMM.

Average surface slope has been achieved from the digital elevation model (DEM) using ArcGIS 9.3 software. Characteristic width of the overland flow path calculated

via equation 1.

$$L = \frac{C\sqrt{A}}{1.128}\left[1 - \sqrt{1 - \left(\frac{1.128}{C}\right)^2}\right] \qquad (1)$$

Where $L$ is the width parameter (m), A is the area of sub watershed (km$^2$) and $C$ is the compactness coefficient. Compactness coefficient calculated via equation 2 for sub watershed with compactness coefficient greater than 1.128. Otherwise, based

on the user manual of SWMM, an initial estimate of the characteristic width is given by the sub watershed area divided by the average maximum overland flow length.

$$C = 0.282 \frac{P}{\sqrt{A}} \qquad (2)$$

Where $P$ is the perimeter of sub watershed (km). Depth of depression storage on impervious and pervious area parameters has been extracted from the values

suggested by ASCE (1992). Manning roughness coefficient was obtained from McCuen et al (1996) and ASCE (1982) manuals. Curve number method was selected for modelling the infiltration process. Land use map of the study area was prepared via processing the Thematic Mapper(TM) images in the IDRISI Selva and ArcGIS 9.3 software. Land use map consisted five class include residential area,

green space, main roads, dense rangeland and degraded rangeland or urban flatted



land. Soil texture of the study watershed achieved from soil surveys deserts atlas of Iran and controlled with soil studies of Agriculture and Natural Resources Research and Education Center of zanjan. Soil hydrological group map was determined based on NRCS Hydrologic Soil Group Definitions in user manual of SWMM (Rossman,

2009). Percent of the impervious area was also calculated based on the land use map of 2012 (Figure 3). Based on the land use map, urban areas, main roads, green space, dense and destroyed rangeland were occupied 82.9, 5.5, 3, 0.4 and 8.2 percent of the city area respectively.

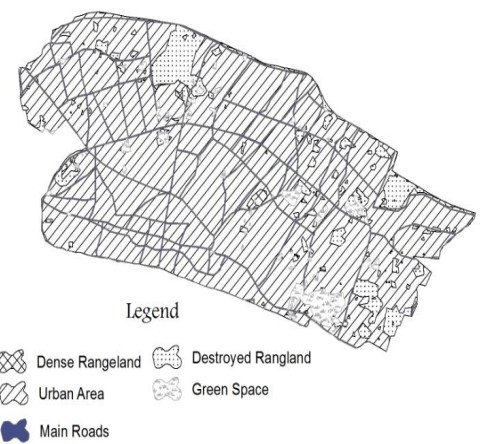

Legend

Dense Rangeland    Destroyed Rangland

Urban Area    Green Space

Main Roads


**Figure 3.** Land use map of the study area

## Rainfall hyetograph generation

Rainfall is a climate parameter of SWMM model that entered to the model in the form of hyetograph. Maximum flood occur when rainfall duration is equal to time of

concentration. In this study, time of concentration for all sub watersheds was computed via TR-55 model suggested by natural resources conservation service (2009). Rainfall hyetographs were prepared using alternating block method. The alternating block method is a simple way for developing a design storm from an IDF curve. The design storm produced by this method specifies the rainfall depth

occurring in "n" successive time intervals of duration ($\Delta t$) over a total duration ($T_d = n * \Delta t$). Based on the design return period, the rainfall intensity extracted from the IDF curve/relation for each of the durations (Butler and John, 2011). This hyetograph represent a rainfall with distinct return period and a rainfall duration equal



or less than $T_d$. Also when rainfall duration is less than $T_d$, the central part of the
main hyetograph with rainfall duration of $T_d$ will be used (Behbahani, 2009).

## Rainfall hyetograph of zanjan city based on Sherman equation

The rainfall IDF curves was derived for all rain gauges stations using an empirical
equations. This equation represent a relationship between maximum rainfall intensity
as dependent variable and other parameters such as rainfall duration and frequency
as independent variables (Le et al 2006). In this study, Sherman equation was used
for Zanjan City (Equation 3)

$$i = \frac{a}{(d+b)^e}$$    (3)

Where i is rainfall intensity (mm/hour); d is duration (minutes); a, b and e are
constant parameters related to the metrological conditions. These empirical
equations show rainfall intensity decreases with rainfall duration for a given return
period. At the Zanjan station, the parameters of Sherman empirical equation were
determined in 1995(Meteorological Organization of I.R IRAN), (Table 2).

**Table 2.** Constant parameter with Sherman empirical equation at the Zanjan City watershed in
different return period

| Return periods T(year) | α | b | e |
|---|---|---|---|
| 2 | 2654.628 | 164.735 | 1.093 |
| 5 | 1977.214 | 87.108 | 1.017 |
| 10 | 2111.948 | 69.215 | 1.007 |
| 20 | 2473.737 | 60.915 | 1.016 |
| 50 | 2884.057 | 53.450 | 1.021 |

The rainfall IDF curves for the Zanjan station was constructed with the Sherman
equation (Figure 4).



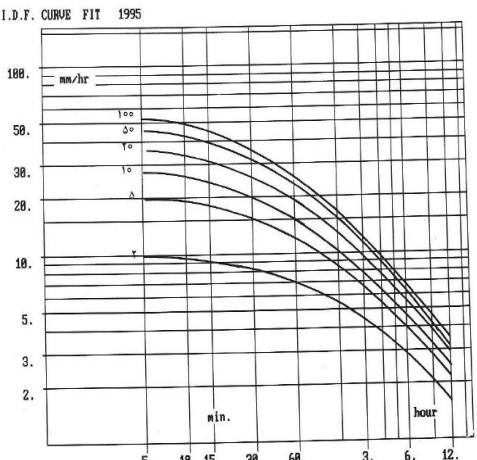


**Figure 4.** Rainfall intensity–duration–frequency curves of Zanjan station for 1995(Meteorological Organization of I.R IRAN, 1955)

Zanjan rainfall hyetographs in different return periods in time intervals of 10, 20, 30 and 40 minute was prepared using rainfall IDF curves of the year 1995.

## Rainfall hyetograph of zanjan city based on Ghahreman and Abkhezr equation

Due to climate change, Ghahreman and Abkhezr (2004) show that rainfall IDF
curves significantly has changed in the recent years. They presented a new equation for indicating relationship between rainfall IDF curves parameters in Iran (Equation 4)

$$R_t^T = At^B[\alpha_1 + \alpha_2 ln(T - \alpha_3)]R_{60}^{10} \qquad (4)$$

Where $R_t^T$ is rainfall depth (mm) with time increment of "t" and return period of T. $A$ and $B$ are the coefficients of rainfall duration (for rainfall less or equal to an hours are
0.1299 and 0.4952 respectively). $\alpha_1$, $\alpha_2$ and $\alpha_3$ are coefficients of rainfall duration (for rainfall less or equal to two hours are 0.4608, 0.2349 and 0.62 respectively). $R_{60}^{10}$ is hourly rainfall with 10-year return period. $R_{60}^{10}$ calculated via equation (5)

$$R_{60}^{10} = e^{0.291}(R_{1440}^2)^{0.694} \qquad (5)$$

Where $R_{1440}^2$ is the average of the maximum daily rainfall that calculated based on
the maximum of daily rainfall from 1969- 2015 in zanjan station. Rainfall hyetographs of the study area for different return periods and rainfall duration (10, 20, 30 and 40 minute) were prepared using equation (4) and (5).





## Model Calibration

Calibration of the SWMM model was proceeded by comparing real field measured hydrographs with simulated flow hydrographs (Zaghloul and Abu Kiefa, 2000). Model calibration is the process of achieving a correspondence between model estimates and field data. For SWMM model calibration, the goal of calibration was to achieve agreement between measured and simulated peak flow rates. The evaluation criteria
of root mean square error (RMSE) was used to compare the simulated model output with the observed data. Root mean square error (RMSE) for discharge is based on equation 6.

$$RMSE = \sqrt{\frac{\sum_{i=1}^{n}[Q_0(i) - Q_s(i)]^2}{n}}$$    (6)

Where $Q_s(i)$ and $Q_o(i)$ are the simulated and observed discharges, respectively,
and, n is number of observations in the time series.

## Results and discussion

Based on both Sherman and Ghahreman and Abkhezr methods, design rainfall hyetograph developed in 10-minute growths for a 50-year return period with 40-
minute duration (Figure 5)

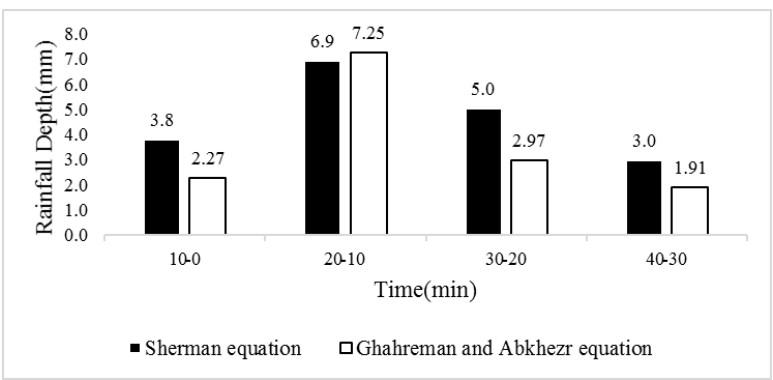

**Figure 5.** A design rainfall hyetograph created via alternative block method based on Sherman
equation, and Ghahreman and Abkhezr equations


Based on Sherman, and Ghahreman and Abkhezr equations, design rainfall hyetograph was developed in 10-minute increments for different return periods with 40-minute duration using alternative block method (Table 3).


**Table 3.** Design rainfall hyetograph developed in 10-minute increments for different return periods using Sherman and Ghahreman and Abkhezr equations

| Method | Return Period(year) | Time(min) | | | | Rainfall Depth(mm) |
|---|---|---|---|---|---|---|
| | | 0-10 | 10-20 | 20-30 | 30-40 | |
| | 2 | 1.2 | 1.6 | 1.4 | 1.1 | 5.3 |
| | 5 | 2.1 | 3.1 | 2.5 | 1.8 | 9.6 |
| Sherman | 10 | 2.7 | 4.3 | 3.3 | 2.2 | 12.5 |
| | 20 | 3.2 | 5.4 | 4.1 | 2.5 | 15.2 |
| | 50 | 3.8 | 6.9 | 5.0 | 3.0 | 18.7 |
| | 2 | 0.9 | 2.8 | 1.2 | 0.7 | 5.6 |
| | 5 | 1.3 | 4.3 | 1.7 | 1.1 | 8.5 |
| Ghahreman and Abkhezr | 10 | 1.6 | 5.2 | 2.1 | 1.4 | 10.3 |
| | 20 | 1.9 | 6.1 | 2.5 | 1.6 | 12.1 |
| | 50 | 2.3 | 7.3 | 3.0 | 1.9 | 14.5 |

According to results, the rainfall depths increased with increasing of return period, whereas rainfall amount decreased with increasing rainfall duration in all return

periods. The results obtained from the two methods have a good uniformity. Based on design hyetographs in different return periods, rainfall depth calculated via Sherman method are greater than rainfall depth calculated via Ghahreman and Abkhezr method except for 2-year return period. As recent data was used in Ghahreman and Abkhezr equation, we can conclude that rainfall depth decreased in

the recent decade (Table 3).

Based on the results, peak of the hyetograph created via Ghahreman and Abkhezr method is greater than Sherman method (Figure 5). Design hyetographs for 50-year return period indicate that 57.22 percent of rainfall depth has been occurred in the first twenty minutes for both Sherman and Ghahreman and Abkhezr methods,

whereas the percent of rainfall depth that happened after hyetograph peak was 42.88 percent in the Sherman method and 33.79 percent in Ghahreman and Abkhezr method (Table 3). This results indicate that estimated hyetograph peak in Ghahreman and Abkhezr method is greater than Sherman method while the depth of rainfall in Sherman method is greater than Ghahreman and Abkhezr.



The results of calibration model based on measured three rainfall runoff events has been shown in Table 4.

**Table 4.** Calibration of the model results with peak flow

| Rainfall- runoff events | Observed peak flow (m³/sec) | Simulated peak flow (m³/sec) | RMSE |
|---|---|---|---|
| 02 May 2016 | 0.088 | 0.09 | 0.005 |
| 03 May 2016 | 0.018 | 0.02 | 0.003 |
| 10 May 2016 | 0.022 | 0.02 | 0.001 |

According to calibration of model results, RMSE criteria imply that the prediction errors are well balanced and simulated hydrograph looks rather reasonable. Comparison has been made between simulated and measured hydrographs (Figures 6a, b and c). The results of calibration has confirmed the accuracy of model simulation.

Based on entered hyetographs characteristics, change in flood properties were simulated via SWMM model. Maximum runoff has been calculated via SWMM model in different return period for different watershed outlets. The results of one outlets indicated in Figure 7.




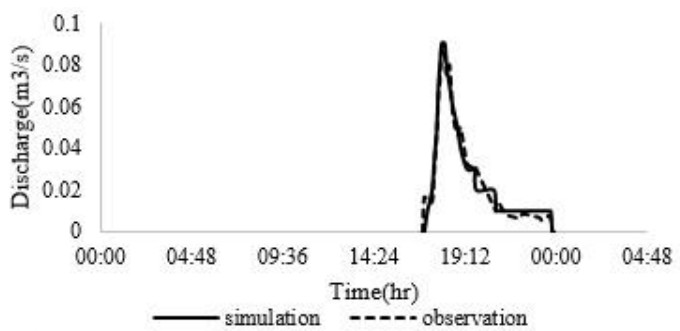

Fig 6a. Calibration outfall hydrograph for rainfall- runoff event at 02 May 2016

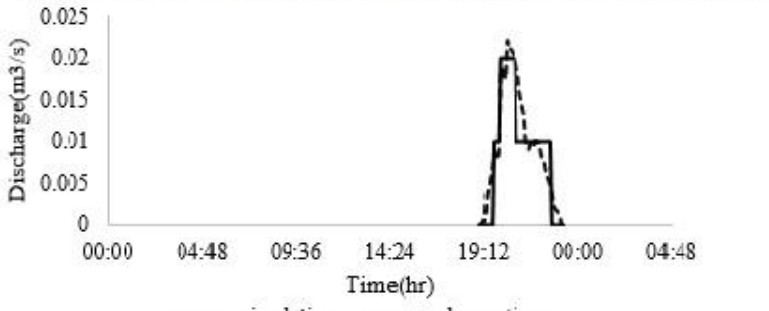

Fig 6b. Calibration outfall hydrograph for rainfall- runoff event at 03 May 2016

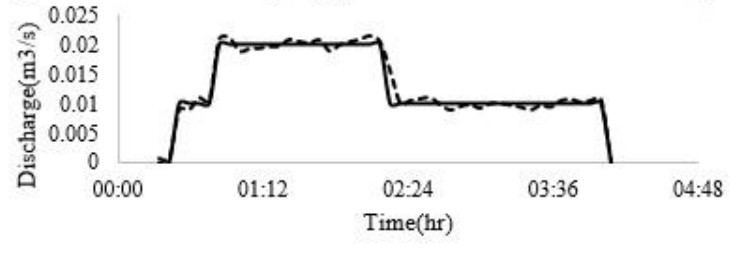

Fig 6c. Calibration outfall hydrograph for rainfall- runoff event at 10 May 2016

**Figure 6.** Calibration outfall hydrographs


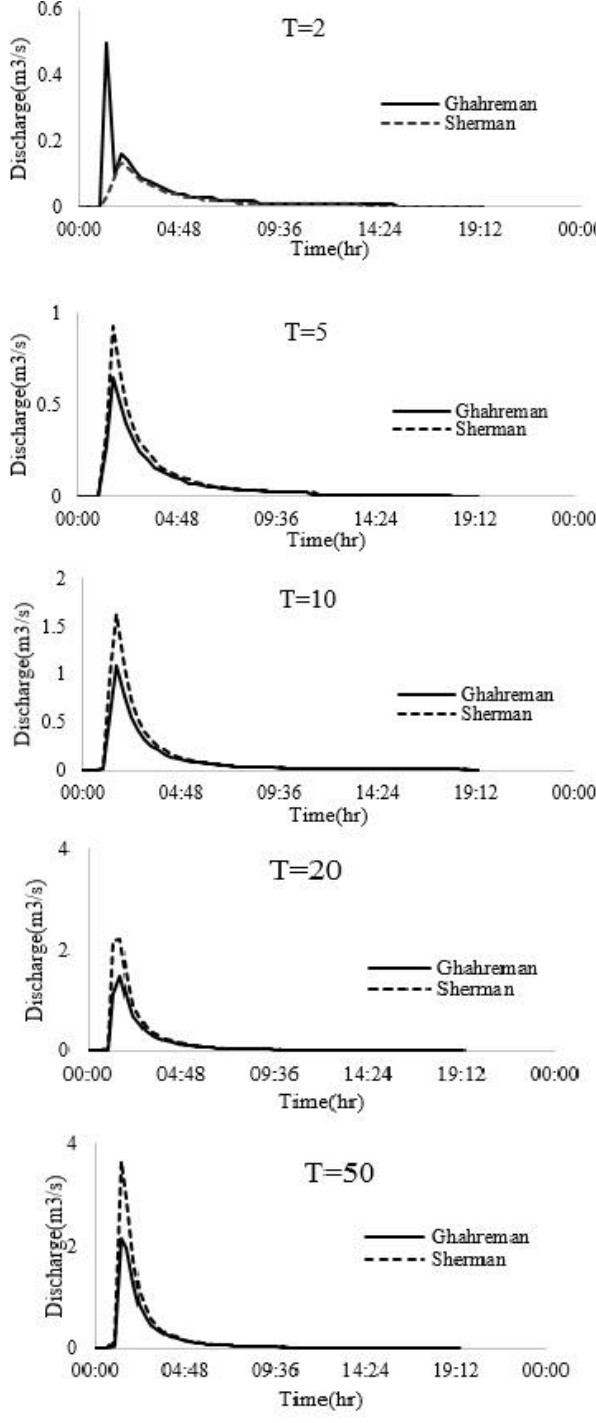


**Figure 7.** The estimated maximum runoff based on two made hyetograph in different return period





The results of estimated maximum runoff based on two made hyetograph in different return period in total of urban drainage system has been shown in Table 5.

**Table 5.** The estimated maximum runoff of urban drainage system based on two made hyetograph in different return period

| Return period | 2 | 5 | 10 | 20 | 50 |
|---|---|---|---|---|---|
| Sherman method | 0.44 | 3.08 | 6.31 | 9.95 | 15.39 |
| Ghahreman and Abkhezr method($m^3$/s) | 0.53 | 2.17 | 3.85 | 5.85 | 8.97 |
| Difference between two method($m^3$/s) | 20 | -30 | -39 | -41 | -42 |

According to results, for 2-year return period, estimated peak runoff has increased by 20 percent using Ghahreman and Abkhezr hyetographs compare to Sherman
method. While for 5, 10, 20 and 50-year return periods, the peak runoff has been decreased by 30, 39, 41 and 42 percent respectively in the Ghahreman and Abkhezr method compare to Sherman method. Table (6) indicate estimated total runoff volume for urban drainage system of zanjan city watershed.

**Table 6.** The estimated total runoff volume of urban drainage system based on two made hyetograph
in different return period

| Return period | 2 | 5 | 10 | 20 | 50 |
|---|---|---|---|---|---|
| Sherman method(m3/s) | 6.9 | 30.5 | 47.7 | 63.3 | 83.4 |
| Ghahreman and Abkhezr method($10^6$ Liter) | 8.5 | 24.2 | 35.2 | 45.6 | 59.5 |
| Difference between two method($10^6$ Liter) | 23 | -21 | -26 | -28 | -29 |

The results show that runoff volume decreased using prepared hyetographs of Ghahreman and Abkhezr method compare to Sherman method except for 2-year return period. Decreasing of total runoff volume for 5, 10, 20 and 50 year return
periods using prepared hyetographs of Ghahreman and Abkhezr method was 21, 26, 28 and 29 percent respectively. While for 2-year return period, evaluated runoff volume increased by 23 percent.

Based on the results, for peak runoff evaluated in 50-year return period using Sherman and Ghahreman and Abkhezr hyetograph, percent of flood that occurred
before of peak runoff were 27 and 22 percent respectively (Figure 7).

According to the results of Sherman method, time to peak is 30 minute and base time of runoff hydrograph is 17 hour and 40 minute for Sherman hyetograph. Same

time to peak was observed for Ghahreman and Abkhezr method, while base time of hydrograph decreased by 20 minute. Time to peak is very important for establishment of flood warning systems and prepare the condition for property protect of humans life and increase the safe lives. In generally, we can conclude that peak and volume of runoff estimation need to update for urban runoff modelling.

## Conclusions

Urbanization and climate change affected local rainfall and intensity. As rainfall characteristics are often used to design urban drainage system, so watershed modelling and estimate of the flood properties, updating and reviewing of rainfall characteristics is necessary. This study was conducted for analyzing the effect of rainfall IDF curves updating on the flood properties in Zanjan city watershed using SWMM model. Design storm in different return period and rainfall duration determined based on two method (Sherman equation, Ghahreman and Abkhezr method). Ghahreman and Abkhezr (2004) attempting to reform the equations of estimating the rainfall intensity in Iran using larger statistic period length. Prepared hyetographs show that Sherman method gave some larger rainfall intensity compared to Ghahreman and Abkhezr method except for 2-year return period. Peak and total runoff volume follow trend of rainfall intensity. Change in rainfall parameters play an important role in designing and managing of the urban runoff. According to results, rainfall intensity and flood volume decreased when Ghahreman and Abkhezr method was used. Same results was reported by Desramaut (2008) that indicated that change in rainfall characteristics lead to runoff decreasing. Willems (2011) also indicated that changes in flood frequencies of sewer systems and overflow frequencies of storage facilities should be quantified based on the climate scenarios and related changes in rainfall statistics. In the study area, as peak and volume of runoff decreased in the recently decade due to climate change, the urban drainage system of Zanjan city watershed has enough transfer capacity against the flood condition.

## Acknowledgments

This work was funded through the University of Kahsn in Iran as a PhD thesis. The authors are grateful to the university for this generous support.



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
