# Peer review of "Modelling and assessment of urban flood hazards based on rainfall intensity-duration-frequency curves reformation"

_Natural Hazards and Earth System Sciences, 2016_

## Referee Comment (RC1) · Anonymous Referee #1 · 13 Nov 2016

The title of the paper indicates a very interesting piece of work. However, I have difficulties following the line of work. Below I list the most important of my concerns.

The authors use the word flooding many times throughout the study but never actually assess flooding, neither theoretically, nor in the case study. This is probably because they use a model for sub-surface piped sewer systems using 1D simulations without capabilities of simulating flooding. They may use the tool for modelling on-ground runoff in channels? Please justify your modelling approach, either by explaining how you have adapted/used the software or by shifting to one of the many 1D2D tool available. Please also choose a suitable title.

According to the authors one of the key objectives is to update and improve previous

work on IDF-relationships for the region. However, in the end I cannot derive if new data were used or not – and what the findings were.

You claim that the model is calibrated and performs well. Your watershed is 39 km2 and your peak discharge is less than 0.1 m3/s in the largest of the three events you have measured. These are very small events that are not representative for the flows you are trying to model. Hence the model is NOT calibrated.

You compare two methods to estimate design storms. I miss a discussion about which method you prefer and why.

There is a very non-linear response between peak precipitation and peak runoff that are not justified based on the manuscript. You have a high degree of urbanization and hence the response should be more linear. Also, there are some things that makes me wonder if the results are realistic at all. The non-linearity could be the result of low degree of non-permeable surfaces in spite of the high degree of urbanization and hence a response from the pervious surfaces. However, if this is the case then the time of concentration should be higher than 40 minutes from a watershed of the size studied. In conclusion it is absolutely impossible to replicate the study even if all data were presented because critical information is missing.

The authors claim that they do the study because they wish to study the impacts of further urbanization and climate change impacts and cite the works by e.g. Willems and Semadeni-Davies. However, the paper contains no attempt to make projections into the future, nor how to manage current deficiencies (if any). By the way, if you wish a more recent and white publication for the work by Willems you can cite the open source publication where Willems is also author (Arnbjerg-Nielsen et al 2013).

It is very difficult to follow the line of thought several places because of poor language. Not even the first sentence in the abstract is proper English.

Other comments:
P2, L63: You cannot use the rational method to determi the flooded urban area directly. The title of the work by Asgari et al also suggests that the statement does not reflect the content of what is cited.

P5, L127: Please present the Sherman equation the first time it is mentioned or give a reference. If you assume it known the first time there is no need to write it out on page 8.

P6, L163: Since you have DEM data available it would be possible to extend the analysis to cover flooding.

P7, L201: Correct citation is to Butler and Davies.

P11, Table 3: The numbers in columns three to six does not have the unit (min). Table should be rearranged.

---

## Referee Comment (RC2) · Anonymous Referee #2 · 17 Nov 2016

[referee-annotated manuscript omitted]

---

## Author Comment (AC1) · 25 Nov 2016

First of all, we should thank very much for your great effort regarding our manuscript. Thank you very much for your great favor regarding our manuscript. The scientific comments and suggestions on the language and structure of the manuscript were really helpful. We have modified the manuscript accordingly, such that the detailed corrections are listed below point by point. Please do not hesitate to inform us regarding any extra comments/considerations

Interactive comment on "Modelling and assessment of urban flood hazards based on rainfall intensity-duration-frequency curves reformation" Referee #1: nhess-2016-304-RC1

Referee #1: The authors use the word flooding many times throughout the study but never actually assess flooding, neither theoretically, nor in the case study. This is probably because they use a model for sub-surface piped sewer systems using 1D simulations without capabilities of simulating flooding. They may use the tool for modelling on-ground runoff in channels? Please justify your modelling approach, either by explaining how you have adapted/used the software or by shifting to one of the many 1D2D tool available. Please also choose a suitable title:

Authors answer: As you know Flooding in urban areas can occurred via flash floods, or coastal floods, or river floods, but there is also a specific flood type that is called urban flooding. Urban flooding is specific in the fact that the cause is a lack of drainage in an urban area. In this study, this kind of flood was investigated. As you write, SWMM is a 1D model that used for single event or long-term (continuous) simulation of runoff quantity and quality from primarily urban areas. The runoff component of SWMM operates on a collection of sub catchment areas that receive precipitation and generate runoff and pollutant loads. The routing portion of SWMM transports this runoff through a system of pipes, channels, storage/treatment devices, pumps, and regulators. This model can also simulate the flow rate and flow volume of sub catchments that arrived to the channels. Based on the flow rate or volume and the dimensions and transfer capacity of channels, model indicate that where nodes or conduit may surcharged or flooded. The objective of this study was to investigate the effects of rainfall IDF curves change on the flood properties (runoff peak and volume). We change paper title as you suggested. "The effects of rainfall intensity-duration-frequency curves reformation on urban flood characteristics in semi-arid environment" As we write in the discussion, in our study area, peak and volume of runoff decreased in the recent years due to climate change, so urban drainage system has enough transfer capacity against the flood condition. It means that based on the results, the urban drainage system of Zanjan city watershed don't have a problem in the terms of flooding or surcharging condition.

Referee #1: According to the authors one of the key objectives is to update and improve previous work on IDF-relationships for the region. However, in the end I cannot derive if new data were used or not – and what the findings were

Authors answer: Old rainfall IDF curves were prepared based on Sherman method using rainfall data of 1972-1993 (P4, L125 - P5, L126). At the first step, design hyetographs of the study area was prepared via this method .This hyetographs was used as the input of SWMM model for estimating peak and volume of runoff. In 2004, rainfall IDF curves were updated by Ghahreman and Abkhezr (2004), using long term rainfall data (1972-2004). A new general relationship for rainfall IDF curves was introduced. According to Ghahreman and Abkhezr method, previous relationship is not useful for estimating 10-year hourly rainfall. At the second step, design hyetographs of the study area was prepared via IDF curve generated via Ghahreman and Abkhezr method .This hyetographs was also used as the input of SWMM model for estimating peak and volume of runoff. This information was added to the paper (page 5 Line 80-90) When we used Ghahreman and Abkhezr method, peak of the rainfall hyetograph increased while the depth of rainfall decreased (Compare to Sherman method). This information was added to the paper ( Conclusion) Referee #1: You claim that the model is calibrated and performs well. Your watershed is 39 km2 and your peak discharge is less than 0.1 m3/s in the largest of the three events you have measured. These are very small events that are not representative for the flows you are trying to model. Hence the model is NOT calibrated

Authors answer: In this study, runoff measurement was done manually via field measurements for three events in one sub basin and the calibrated model was used for the study watershed. The area of this sub basin was only 4.6 km2. So peak discharge of this sub-basin was 0.1 m3/s. Furthermore, calibration files normally contain measurements of only a single parameter at one locations that compared with simulated values in Time Series Plots. At this research, in order to enhance the accuracy of the model calibration, we register three calibration data (Link flow velocity, link flow depth and link

flow rate). So, flow rate in any measured runoff event, flow velocity and flow depth were compared with simulated runoff velocity and depth. This explanation was added to the text ( Page11, Line190-195)

Referee #1: You compare two methods to estimate design storms. I miss a discussion about which method you prefer and why?

Authors answer: The main objective of this study was to compare the flood properties (flow rate and volume) in two time steps. According to results, more accuracy was observed between simulated and real condition when Ghahreman and Abkhezr method was used. When we used Ghahreman and Abkhezr method, peak of the rainfall hyetograph increased while the depth of rainfall decreased (Compare to Sherman method). This dissection was added to the text

Referee #1: There is a very non-linear response between peak precipitation and peak runoff that are not justified based on the manuscript. You have a high degree of urbanization and hence the response should be more linear. Also, there are some things that makes me wonder if the results are realistic at all. The non-linearity could be the result of low degree of non-permeable surfaces in spite of the high degree of urbanization and hence a response from the previous surfaces. However, if this is the case then the time of concentration should be higher than 40 minutes from a watershed of the size studied. In conclusion it is absolutely impossible to replicate the study even if all data were presented because critical information is missing.

As we wrote, this urban watershed has 16 sub-basin (Table 1). Hyetograph of each hydrological unit was prepared separately (16 hyetograph based on Sherman method and 16 hyetograph based on Ghahreman and Abkhezr) and presented to model. For each outlet, a separated hydrograph were created via SWMM model. Estimated maximum runoff of one sub- basin (Sub-basin number 16) was indicated in figure 7. The presented time of concentration (40min) is also time of concentration of this sub basin. We have an acceptable agreement between peak precipitation and peak runoff .In this

sub-basin, based on the design hyetographs, for 40 minute precipitation, peak precipitation has been occurred in the first twenty minutes for both Sherman and Ghahreman and Abkhezr methods and peak runoff was occurred after 30min ( with 10 minute delay) . Maximum flow (peak runoff) and maximum runoff volume for urban watershed was calculated as the sum of the sub-basins outlet. Table 5 indicate the estimated maximum runoff of urban drainage system based on two made hyetograph in different return period for total of the urban watershed drainage system (sum of the 16 sub basin). We didn't present a single time of concentration or hydrograph for studied watershed due to several out let of the watershed (Figure 2). explanation was added to the text

Authors answer:

Referee #1: The authors claim that they do the study because they wish to study the impacts of further urbanization and climate change impacts and cite the works by e.g. Willems and Semadeni-Davies. However, the paper contains no attempt to make projections into the future, nor how to manage current deficiencies (if any). By the way, if you wish a more recent and white publication for the work by Willems you can cite the open source publication where Willems is also author (Arnbjerg-Nielsen et al 2013).

Authors answer:

In this study, the effect of climate change on urban runoff was investigated considering the effects of climate change on rainfall properties (Rainfall hyetograph). According to results, rainfall distribution pattern was changed when recent rainfall data used for preparation of hydrograph (via climate change), consequently urban runoff characteristics were changed. This paragraph was added to conclusion Both Semadeni-Davies et al., 2008 and Willems, 2011, investigated the effects of climate change (rainfall properties) on the runoff generation (flooding conditions). So in this paper, we cited these works because they were related to main objective.

Referee #1: It is very difficult to follow the line of thought several places because of

poor language. Not even the first sentence in the abstract is proper English.

Authors answer: We edit our manuscript for a better language condition. Referee #1: P2, L63: You cannot use the rational method to determine the flooded urban area directly. The title of the work by Asgari et al also suggests that the statement does not reflect the content of what is cited

Authors answer: This sentence was improved.

Referee #1: P5, L127: Please present the Sherman equation the first time it is mentioned or give a reference. If you assume it known the first time there is no need to write it out on page 8. Authors answer: The Sherman equation was presented at the first time it is mentioned.

Referee #1: P6, L163: Since you have DEM data available it would be possible to extend the analysis to cover flooding.

Authors answer: As we explain in the first comment, Storm water management model is a 1D model that cannot do flood zoning simulation . For analysis cover flooding, in addition to DEM data, the Geometry data, Flow data, and Plan data are also need.

Referee #1: P7, L201: Correct citation is to Butler and Davies.

Authors answer: What you think is right and in the final files of this paper will be edited and modified.

Referee #1: P11, Table 3: The numbers in columns three to six does not have the unit (min). Table should be rearranged.

Authors answer: Table was rearranged as below

Table 3. Design rainfall hyetograph developed in 10-minute increments for different return periods using Sherman and Ghahreman and Abkhezr equations Method Return Period(year) Rainfall Incremental Depth(mm) Time(min) Rainfall Depth(mm) 0-10 10-20 20-30 30-40 Sherman 2 Incremental Depth(mm) 1.2 1.6 1.4 1.1 5.3 5

2.1 3.1 2.5 1.8 9.6 10 2.7 4.3 3.3 2.2 12.5 20 3.2 5.4 4.1 2.5 15.2 50 3.8 6.9 5.0 3.0 18.7 Ghahreman and Abkhezr 2 Incremental Depth(mm) 0.9 2.8 1.2 0.7 5.6 5 1.3 4.3 1.7 1.1 8.5 10 1.6 5.2 2.1 1.4 10.3 20 1.9 6.1 2.5 1.6 12.1 50 2.3 7.3 3.0 1.9 14.5

Please also note the supplement to this comment:
http://www.nat-hazards-earth-syst-sci-discuss.net/nhess-2016-304/nhess-2016-304-AC1-supplement.pdf

**Supplement:**

**The effects of rainfall intensity-duration-frequency curves reformation on urban flood characteristics in semi-arid environment**

Reza Ghazavi[1*], Ali Moafi Rabori[2] and M. Ahadnejad Reveshty[3]

[1.] *Associate Professor, Department of Watershed Management, Faculty of Natural Resources and Earth Sciences, University of Kashan, Kashan, Iran*

[2.] *PhD Candidate of Watershed Management, Department of Watershed Management, Faculty of Natural Resources and Earth Sciences, University of Kashan, Kashan, Iran*

[3.] *Associate Professor, Department of Geography, Faculty of Humanities, University of Zanjan, Iran*

*Corresponding author: Reza Ghazavi (*ghazavi@kashanu.ac.ir*),*

**Abstract**

A design storm is a theoretical storm event based on rainfall intensities associated with frequency of occurrence and having a set duration. Estimating design storm via rainfall intensity–duration–frequency (IDF) curves is important for hydrological planning of urban areas. The main aim of this study was to estimate impacts of rainfall IDF curves change in flood properties of an urban area using Storm water Management Model (SWMM). IDF curves of Zanjan city located in the north-west of Iran were generated by Sherman method and Ghahreman and Abkhezr method. Sherman empirical equation was determined in 1995, but due to climate change, Ghahreman and Abkhezr (2004) show that rainfall IDF curves has changed significantly in the recent years. They presented a new equation to indicate the relationship between rainfall IDF curve parameters in Iran. The accuracy of model simulations was confirmed based on the results of calibration. According to results, estimated rainfall depth for different return periods was decreased in the recent years except for 2-year return period, consequently runoff peak was decreased. Decrease of runoff peak was 30, 39, 41 and 42 percent for 5-10-20 and 50-year return periods respectively.

**Key words:** Design storm, Drainage system, Flood, Rainfall IDF curve, Stormwater, SWMM, Urban, Zanjan city

**Introduction**

Due to variation and complexity of land use, population and social economic activities in urban area, storm water runoff management is a complex task in such area (Choi and Ball, 2002; Hoang et al., 2016). This issue will become more complex due to urban development. By 2030, the urban population will reach 5 billion or 60 percent of the world's population (UN, 2006). In many countries, less than 5 percent of land occupied via urban area, consequently, concentration of human activities, shortages and unavailability of resources intensifies local competition for all types of resources, with water amongst the most vital (Zoppou, 2001). Due to local change in hydrological cycle and hydro-meteorological conditions in urban areas, urbanization should increase flood risk (Huong and Pathirana, 2013; Ahilan et al., 2016; Eunsek et al., 2016). Modeling is important for facilitating the development of urban drainage infrastructure design and planning (Choi and Ball, 2002). Estimate and collection of input parameters (measured and inferred) is very important in the catchment modelling. Rainfall intensity-duration-frequency (IDF) curves are important parameters in hydrological modelling. Today, due to urban development, urban flooding increased in terms of intensity and frequency.

Various tools such as structures or non- structures used for urban flooding management and control. Estimating the design storm has an important role in designing and operations of structural hard-engineered solutions for urban runoff management. Use of rainfall IDF curves is a critical method for estimating the design storm. Modeling the impacts of urbanization and climate change on flood properties shows that increase in rainfall intensity and impervious surfaces should cause flashier runoff periods, greater peak flows and heightened risk of flooding (Semadeni-Davies et al., 2008). Several studies indicate that rainfall condition changed due to climate change (Watt et al., 2003; Ghahreman and Abkhezr, 2004). These changes should lead to change in rainfall IDF curves (Ghahreman and Abkhezr, 2004). The impact of the current and future climate change on the rainfall IDF curves and urban design storms in Quebec was estimated using SWMM model (Desramaut, 2008). For developing rainfall intensity-duration-frequency curves in scarce data region, in North-West of Angola, an index flood procedure was used for generate the theoretical regional distribution equation (Ayman et al., 2011).

In order to design the drainage structures, Ibrahim (2012) conducted a study for developing rainfall IDF relationship for two regions in Saudi Arabia. Using improved IDF relations in Khorasan region of Iran, a study was performed to determine the spatial distribution of storms (Akbari et al., 2014).

50 During the last century, the concentration of greenhouse gases has increased due to urban development and increasing of industrial activities (Prodanovic and Simonovic, 2007). This change can lead to changes in temperature and precipitation characteristics, consequently urban flood characteristics should change. In this study, rainfall IDF curves were prepared based on two methods: Sherman method using rainfall data of 1972-1993; Ghahreman and Abkhezr method using long term rainfall data (1972-2004). Then the effects of rainfall intensity-duration-frequency curves reformation on urban flood

55 characteristics investigated using SWMM model. Finally the effects of rainfall IDF curves updating on peak and volume of flood was examined.

**Study area**

The study area is located in the center of Zanjan province, north-west of Iran (latitude 36° 38′ 26″ and 36° 42′ 20″N, longitude 48° 26′ 29″ and 48° 35′ 02″ E). Altitude of the study area range from 1590 m above mean sea level in the southern

60 plain to 1773m in the northern mountain (Fig.1). Total area of the study urban watershed is about 39 $km^2$ and the mean annual rainfall is 290 mm. The main part of rainfall in the study area occurred in the autumn and spring. Urban runoff drain into Zanjanrood River via several artificial canals. Flow direction of this canals is from north to south of urban area. Gavazaang earth dam has been built at the north of the city. This dam limit upstream surface water and floods. The study area includes central business district of Zanjan city, public parks, green space, residences and streets. This city experienced

65 rapid development and population expansion during 1956-2012.

[Figure]

**Fig. 1**. Locations of the Zanjan City Watershed

**Material and Methods**

70   Flooding in urban areas can occurred via river floods, coastal floods or flash floods, but there is also a specific flood type that is called urban flooding. Urban flooding is specific in the fact that the cause is a lack of drainage in an urban area. In this study, this kind of flood was investigated using IDF curves and SWMM model.

**Model description**

75   The EPA Storm Water Management Model (SWMM) developed under the support of the US Environmental Protection Agency (Huber and Dickinson, 1992). SWMM is a dynamic rainfall-runoff simulation model that computes runoff quantity from primarily urban areas. SWMM widely used throughout the world for planning, analysis, and designing related to stormwater runoff, combined sewers, sanitary sewers, and other drainage systems in urban and non-urban areas. The runoff component of SWMM operates on a collection of sub catchment areas that receives precipitation and generates runoff and

80   pollutant loads. The routing portion of SWMM transports this runoff through a system of pipes, channels, storage/treatment devices, pumps and regulators. SWMM tracks the quantity of runoff generated within each sub catchment and the flow

rate, flow depth, and quality of water in each pipe and channel during a simulation period comprised of multiple time steps (Gironas et al., 2009).

**Model implementation**

The primary objective of this study was to evaluate the hydrologic and hydraulic response of an urban watershed to the rainfall IDF curves updating based on increasing of the statistical period length of the rainfall data.

IDF curves of the study area  were prepared based on Sherman method using rainfall data of 1972-1993.At the first step, design hyetographs of the study area was prepared via this method .This hyetographs was used as the input of SWMM model for estimating peak and volume of runoff.

In 2004, rainfall IDF curves were updated by Ghahreman and Abkhezr (2004), using long term rainfall data (1972-2004). A new general relationship for rainfall IDF curves was introduced.  According to Ghahreman and Abkhezr method, previous relationship is not useful for estimating 10-year hourly rainfall.

At the second step of this study, design hyetographs of the study area was prepared via IDF curve generated via Ghahreman and Abkhezr method .This hyetographs was also used as the input of SWMM model for estimating peak and volume of runoff.

[revised manuscript text omitted]

The rainfall IDF curves for the Zanjan station constructed using Sherman equation (Fig. 4). Rrainfall hyetographs of all studied sub-watersheds in different return periods were prepared in time intervals of 10 minute using rainfall IDF curves of the year 1995.

[Figure]

**Fig. 4.** Rainfall intensity–duration–frequency curves of Zanjan station for 1995(Meteorological Organization of I.R IRAN, 1955)

170

**Rainfall hyetograph of Zanjan city based on Ghahreman and Abkhezr equation**

Due to climate change, Ghahreman and Abkhezr (2004) show that in the recent years, rainfall IDF curves changed significantly. They presented a new equation for indicating relationship between rainfall IDF curves parameters in Iran

175 (Equation 4)

$$(4) \qquad R_t^T = At^B[\alpha_1 + \alpha_2 ln(T - \alpha_3)]R_{60}^{10}$$

Where $R_t^T$ is rainfall depth (mm) with time increment of "t" and return period of T. $A$ and $B$ are the coefficients of rainfall duration (for rainfall less or equal to an hours are 0.1299 and 0.4952 respectively). $\alpha_1$, $\alpha_2$ and $\alpha_3$ are coefficients of rainfall duration (for rainfall less or equal to two hours are 0.4608, 0.2349 and 0.62 respectively). $R_{60}^{10}$ is the hourly rainfall with

180 10-year return period. $R_{60}^{10}$ calculated via equation (5)

$$(5) \qquad R_{60}^{10} = e^{0.291}(R_{1440}^2)^{0.694}$$

Where $R^2_{1440}$ is the average of the maximum daily rainfall that calculated based on the maximum of daily rainfall from 1969- 2015 in Zanjan station. Rainfall hyetographs of all sub watersheds were prepared for different return periods and rainfall duration with 10 minute interval using equation (4) and (5).

Urban study watershed has 16 sub-basin (Table 1). Hyetograph of each hydrological unit was prepared separately (16 hyetograph based on Sherman method and 16 hyetograph based on Ghahreman and Abkhezr) and presented to model. For each outlet, a separated hydrograph were created via SWMM model.

**Model Calibration**

Calibration of the SWMM model was proceeded by comparing real field measured hydrographs with simulated flow hydrographs (Zaghloul and Abu Kiefa, 2000). Model calibration is the process of achieving a correspondence between model estimates and field data. For SWMM model calibration, the goal of calibration was to achieve agreement between measured and simulated peak flow rates. The evaluation criteria of root mean square error (RMSE) was used to compare the simulated model output with the observed data. Root mean square error (RMSE) for discharge is based on equation 6.

(6) $$RMSE = \sqrt{\frac{\sum_{i=1}^{n}[Q_0(i)-Q_s(i)]^2}{n}}$$

Where Qs (i) and Qo (i) are the simulated and observed discharges, respectively, and, n is number of observations in the time series.

In this study, for model calibration, rainfall and runoff measured in 10 minute interval for three events in one sub basin. The area of this sub basin was 4.6 km². As all sub-watershed have same rainfall and land use condition, the calibrated model was used for the study watershed. Furthermore, calibration files normally contain measurements of only a single parameter at one locations that compared with simulated values in Time Series Plots. At this research, in order to enhance the accuracy of the model calibration, we register three calibration data (Link flow velocity, link flow depth and link flow rate). So, flow rate in any measured runoff event, flow velocity and flow depth were compared with simulated runoff velocity and depth.

**Results and discussion**

Based on both Sherman and Ghahreman and Abkhezr methods, for all 16 sub-watershed, design rainfall hyetograph developed in 10-minute growths for different return periods. As the same results obtained for all study sub watershed, the results obtained for sub watershed 16 (the biggest sub watershed) was presented in this section. Figure 5 indicates design rainfall hyetograph created via alternative block method based on Sherman equation, and Ghahreman and Abkhezr equations for sub watershed number 16 (Fig. 5)

[Figure]

**Fig. 5.** A design rainfall hyetograph created for sub watershed number 16 via alternative block method based on Sherman equation, and Ghahreman and Abkhezr equations

Based on Sherman, and Ghahreman and Abkhezr equations, design rainfall hyetograph was developed in 10-minute increments for different return periods for all 16 sub watershed using alternative block method. Table 3 indicates design rainfall hyetograph of sub watershed 16.

225 **Table 3.** Design rainfall hyetograph developed in 10-minute increments for different return periods using Sherman and Ghahreman and Abkhezr equations (sub watershed 16)

[revised manuscript text omitted]

**280 Conclusions**

Urbanization and climate change affected local rainfall intensity. As rainfall characteristics are often used to design urban drainage system, so for watershed modelling and estimate of flood properties, updating and reviewing of rainfall characteristics is necessary.

The first step in many hydrological design projects is to determine the maximum rainfall event. IDF curves, which relate

285    the rainfall intensity, duration, and frequency, is the most common method of determining the design storm event. It also provides a summary of the site's rainfall characteristics by relating storm duration and exceedance probability to rainfall intensity which is assumed to be constant over the duration (time of concentration) .The IDF curves of the study area were developed in this study using historical rainfall data available. This study was conducted for analyzing the effect of rainfall IDF curves updating on the flood properties in Zanjan city watershed using SWMM model. Design storm in different return

290    period and rainfall duration determined based on two method (Sherman equation, Ghahreman and Abkhezr method). Ghahreman and Abkhezr (2004) attempting to reform the equations of the rainfall intensity estimating in Iran using larger statistic period length. Prepared hyetographs show that Sherman method gave larger rainfall intensity compared to Ghahreman and Abkhezr method. Estimated peak and total runoff volume follow trend of rainfall intensity.

Rainfall intensity in the IDF Curve is the average rainfall depth that falls per specific time duration. AS Ghahreman and

295    Abkhezr method use longer and newer rainfall data for creating IDF curves, we can conclude that climate change cause change in rainfall characteristics.   According to results, more accuracy was observed between simulated and real condition when Ghahreman and Abkhezr method was used. When we used Ghahreman and Abkhezr method, peak of the rainfall hyetograph increased but depth of rainfall decreased, consequently flood volume decreased. This mean that climate change

would be affected rainfall pattern of the study area. Desramaut (2008) indicated that change in rainfall characteristics lead to runoff decreasing. Willems (2011) also indicated that changes in flood frequencies of sewer systems and overflow frequencies of storage facilities should be quantified based on the climate scenarios and related changes in rainfall statistics. Due to climate change peak and volume of runoff decreased in the recently decade. According to results of SWMM model, the urban drainage system of Zanjan city watershed has enough transfer capacity against the flood condition. But survey information indicated several inundations in some area of the studied watershed. Poor maintenance of drainage systems, instantaneous heavy rainfall, erosion and sedimentation are some parameters that could lead to such temporal inundation.

**Acknowledgments**

[revised manuscript text omitted]

---

## Author Comment (AC2) · 29 Nov 2016

Referee #2: nhess-2016-304-RC2 First of all, we should thank very much for your great effort regarding our manuscript. Thank you very much for your great favor regarding our manuscript. The scientific comments and suggestions on the language and structure of the manuscript were really helpful. We have modified the manuscript accordingly, such that the detailed corrections are listed below point by point. Please do not hesitate to inform us regarding any extra comments/considerations

Referee #2: This research paper attempts to model and evaluate urban flood hazards in a region of Iran. The paper is difficult to read because of the poor English language used by the authors and very weak since several research aspects need to be better

stated and analyzed. Authors answer: We edit our manuscript for a better language condition. All highlighted sentences and punctuated errors were modified.

Referee #2: The general objectives of the work are stated at the end of the introduction and they are not well discussed. Authors answer: This sentences edited at the end of introduction. Also discussion and conclusion parts of paper edited.

Referee #2: The authors want to compare the impact of the use of two IDF curves on the urban drainage system. At the beginning of the work they declare to have derived a new IDF curve but in the central part of the manuscript also this second IDF curve is provided by other studies. Authors answer: The objective of this study was to investigate the effects of rainfall IDF curves change on the flood properties (runoff peak and volume). Two type of hydrograph were used: Sherman method proposed based on rainfall data of 1972-1993. At the first time, this hyetographs used as the input of SWMM model for estimating peak and volume of runoff. In 2004, rainfall IDF curves were updated by Ghahreman and Abkhezr (2004), using long term rainfall data (1972-2004), so new climate condition was considered. A new general relationship for rainfall IDF curves was introduced. At the second time, design hyetographs of the study area was prepared via rainfall IDF curve generated via Ghahreman and Abkhezr method. This hyetographs was also used as the input of SWMM model for estimating peak and volume of runoff. We concluded that change in estimated runoff related to climate change.

Referee #2: Please, see attached the annotated file with further comments and suggested edits. In my opinion the paper must be deeply revised before to be reviewed again. Highlighted sentences require revision. Authors answer: The manuscript edited based on further comments and suggested edits in attached file. Also highlighted sentences and punctuated errors were modified.

Referee #2: In P 8, L 205, Referee wrote so, which is the value of the concentration time? And also referee wrote in P9, L 230, Zanjan rainfall hyetographs in different

return periods in time intervals of 10, 20, 30 and 40 minute was prepared using rainfall IDF curves of the year 1995. Referee wrote why this values (i.e. up to 40 min? is this time of concentration?) Authors answer: As you know maximum flood occur when rainfall duration is equal to time of concentration. Since this urban watershed has 16 sub-basin (Table 1) with different time of concentration, so hyetograph of each sub watershed was prepared separately (16 hyetograph based on Sherman method and 16 hyetograph based on Ghahreman and Abkhezr) based on time of concentration of each sub watershed. Estimated maximum runoff of one sub-basin (Sub-basin number 16) was indicated in figure 7. The presented time of concentration (40min) is also time of concentration of this sub basin.

Referee #2: in P 9, L 234. Referee wrote how about statistical methods??? Where does equation 5 come from? And hourly properties from daily statistics? Authors answer:

These synthetic equations proposed by Sherman to generate rainfall IDF curves and hourly rainfall with 10-year return period in Iran. Rainfall data of 66 rain gauge stations extracted and analyzed for preparing rainfall IDF curves. The Equation 5 is an empirical equation proposed after analyzing the data of Maximum of daily precipitation, average of annual precipitation and the number of rainy days After that, Ghahreman and Abkhezr found that the maximum of daily precipitation and average of annual precipitation had significant effects on estimating the hourly rainfall with 10-year return period. As mentioned in the text, $R\_1440^2$ is the average of the maximum daily rainfall. The amount of $R\_1440^2$ calculated based on the average of maximum of daily precipitation data from 1969- 2015 in Zanjan station.

Referee #2: in the resulting of model calibration (P12, L 295 -305) referee wrote how about the initial conditions???? Which is the operational temporal scale? 10 min? Authors answer: In this study, for model calibration, rainfall and runoff measured in 10 minute interval for three events in one sub basin. The area of this sub basin was 4.6 km2. As all sub-watershed have same rainfall and land use condition, the calibrated

model was used for the study watershed. This explanation added to text

Referee #2: in P 16 in the conclusion referee wrote which is the contribution of this work??

We improve this paragraph.

Base on both reviewer comments, we improve this manuscript. Please let we know for any extra information or comment.

Please also note the supplement to this comment:
http://www.nat-hazards-earth-syst-sci-discuss.net/nhess-2016-304/nhess-2016-304-AC2-supplement.pdf